# NEURAL NETWORK DIFFUSION

## ABSTRACT

Diffusion models have achieved remarkable success in image and video generation. In this work, we demonstrate that diffusion models can also *generate high-performing neural network parameters*. Our approach is simple, utilizing an autoencoder and a standard latent diffusion model. The autoencoder extracts the latent representation of trained model parameters. A diffusion model is then trained to synthesize these latent parameter representations from random noise. It then generates new representations that are passed through the autoencoder's decoder, whose outputs are ready to use as new sets of network parameters. Across various tasks and datasets, our diffusion process consistently generates models of comparable or improved performance over SGD-trained models, with minimal additional cost. Our results encourage more exploration on the versatile use of diffusion models.

## 1 INTRODUCTION

The origin of diffusion models can be traced back to non-equilibrium thermodynamics (Jarzynski, 1997; Sohl-Dickstein et al., 2015), constituting a category of generative models based on probabilistic likelihood. In 2015, Sohl-Dickstein et al. (2015) was the first to utilize diffusion processes to progressively remove noise from inputs and generate clear images. After that, DDPM (Ho et al., 2020), DDIM (Song et al., 2021), and subsistent works (see Sec. 5) refine the framework of diffusion models, employing a training paradigm characterized by integrated forward and reverse processes.

At that time, the quality of images generated by diffusion models had not yet reached the desired level. Guided-Diffusion (Dhariwal & Nichol, 2021) conducts a series of ablations and finds a better architecture, which represents the pioneering effort to elevate diffusion models beyond GAN-based methods (Zhu et al., 2017; Isola et al., 2017) in terms of image quality. Subsequently, GLIDE (Nichol et al., 2021), Imagen (Saharia et al., 2022), DALL·E 2 (Ramesh et al., 2022), and Stable Diffusion (Rombach et al., 2022) achieve notable achievements and promote the application of diffusion models in image and video generation.

Despite the great success of diffusion models in visual generation, their potential in other domains remains relatively underexplored. In this work, we demonstrate the surprising capability of diffusion models in *generating high-performing model parameters*, a task fundamentally distinct from traditional visual generation. Parameter generation is centered around creating neural network parameters, prioritizing numerical properties over visual realism. Meanwhile, it does not have as many successful precedents as visual generation, which leads to challenges for neural network diffusion.

The diffusion-based image generation shares commonalities with the SGD learning process in the following aspects (illustrated in Fig. 1): i) high-quality images and high-performing parameters can be degraded into simple distributions, such as Gaussian distribution, through multiple noise additions. ii) Both neural network training and the reverse process of diffusion models can be regarded as transitions that involve moving from random noise/initialization to specific distributions.

Based on the observations above, we introduce Neural Network Diffusion (DiffNet), a novel approach for parameter updating, which employs diffusion steps to generate high-performing parameters. Our method is simple, comprising an autoencoder and a standard diffusion model. First, for a set of SGD-trained models, the autoencoder is utilized to extract the latent representations for these trained models. Then, we leverage the standard diffusion model to synthesize these latent parameter representations from random noise. Finally, the diffused latent parameter representations are passed through the trained autoencoder's decoder to yield novel and high-performing model parameters. Our

Figure 1: *The left:* illustrates the standard diffusion process on image generation. *The right:* denotes the Batch Normalization (BN) parameter distributions during the training CIFAR-100 with ResNet-18. *The top half of the bracket:* BN weights. *The lower half of the bracket:* BN biases.

approach aims to learn the distribution of high-performing parameters. That is supported by the fact that the diffusion model has the capability to transform a given random distribution to a specific one.

DiffNet exhibits the following characteristics: i). We learn the distribution from a set of trained models and progressively generate high-performing parameters, offering the potential to achieve lossless, even enhanced performance than input models. Compared to the SGD-trained models, on ResNet-18, ViT-Tiny, and ConvNeXt-T, the improvements are up to 0.6, 0.4, and 0.5 across eight datasets, respectively. ii). Our generated models have greater diversity than adding noise and fine-tuning (see it in Fig. 3), which illustrates our approach learns a wide range of high-performing parameter distributions. ii). DiffNet facilitates the rapid generation of a variety of high-performing models. As illustrated in Fig. 5, DiffNet can generate 100 ResNet-18 models within 40 seconds, while SGD would take 22 hours (800s×100) to obtain 100 models. We hope our approach can provide fresh insights into expanding the applications of diffusion models and inspire further research.

## 2 NERUAL NETWORK DIFFUSION

### 2.1 PRELIMINARIES OF DIFFUSION MODELS

Diffusion models typically consist of forward and reverse processes in a multi-step chain indexed by timesteps. We introduce these two processes in the following.

**Forward process.** Given a real image $x_0 \sim q(x)$, the forward process progressively adds Gaussian noise for $T$ steps and obtain $x_1, x_2, \cdots, x_T$. The formulation of it can be written as follows,

$$q(x_t|x_{t-1}) = \mathcal{N}(x_t; \sqrt{1 - \beta_t}x_{t-1}, \beta_t\mathbf{I}), \qquad q(x_{1:T}|x_0) = \prod_{t=1}^{T} q(x_t|x_{t-1}), \qquad (1)$$

where $q$ denotes the forward process, $\mathcal{N}$ represents the operation of adding Gaussian noise that parameterized by $\beta_t$, and $\mathbf{I}$ is the identity matrix. Since only Gaussian noise is added through the chain, there are no trainable parameters in this process. Generally, $T \to \infty$, $x_T$ is full of noise.

**Reverse process.** Different from the forward process, the reverse process aims to train a denoising network to recursively remove the noise from $x_t$. It moves backward on the multi-step chain as t decreases from $T$ to 0. Mathematically, the reverse process can be formulated as follows,

$$p_\theta(x_{t-1}|x_t) = \mathcal{N}(x_{t-1}; \mu_\theta(x_t, t), \Sigma_\theta(x_t, t)), \qquad p_\theta(x_{0:T}) = p(x_T)\prod_{t=1}^{T} p_\theta(x_{t-1}|x_t), \quad (2)$$

where $p$ denotes the reverse process, $\mu_\theta(x_t, t)$ and $\Sigma_\theta(x_t, t))$ are the Gaussian mean and variance that estimated by the denoising network parameter $\theta$. The denoising network is optimized by the standard negative log-likelihood:

$$\mathrm{L}_{dm} = \mathcal{D}_{KL}(q(x_{t-1}|x_t, x_0)||p_\theta(x_{t-1}|x_t)), \qquad (3)$$

where the $\mathcal{D}_{KL}(\cdot||\cdot)$ denotes the Kullback–Leibler (KL) divergence and computes the difference between two distributions.

**Training and Inference procedures.** The goal of the training diffusion model is to find the reverse transitions that maximize the likelihood of the forward transitions in each time step $t$. In practice,

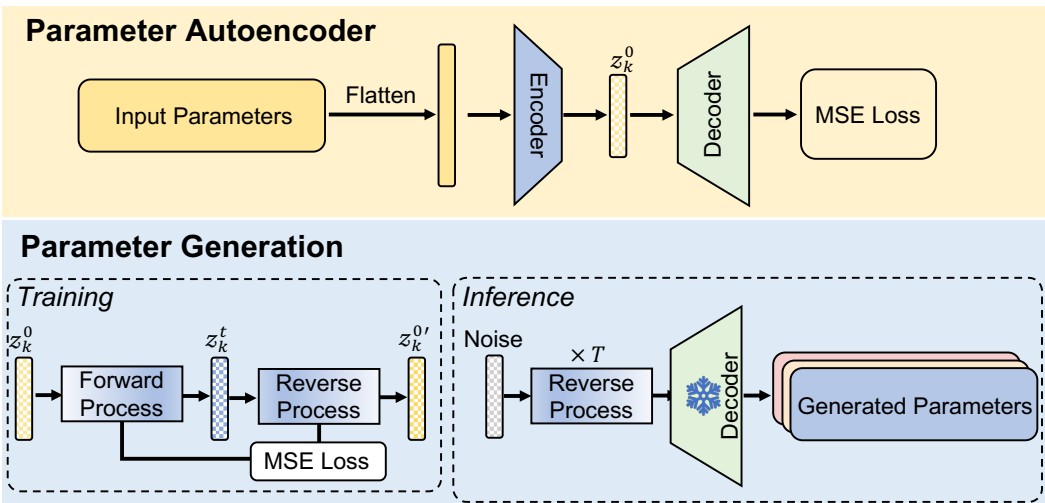

Figure 2: Illustration of the proposed DiffNet framework. DiffNet consists of two processes, namely parameter autoencoder and generation. Parameter autoencoder aims to extract the latent representations that can generate high-performing model parameters via the decoder. The extracted representations are fed into a diffusion model. $z_k^t$ represents the latent representation of the $k$-th input parameter at timestep $t$. During the inference, we freeze the parameters of the autoencoder's decoder.

training equivalently consists of minimizing the variational upper bound. The inference procedure aims to generate novel samples from random noise via the optimized denoising parameters $\theta^*$ and the multi-step chains in the reverse process.

## 2.2 OVERVIEW

We propose neural network diffusion, termed DiffNet, which aims to generate high-performing model parameters from random noise. As illustrated in Fig. 2, our method consists of two modules, namely parameter autoencoder and parameter generation. Given a set of trained high-performing models, we first flatten them into 1-dimensional vectors. Subsequently, we introduce an encoder to extract latent representations from these parameters, accompanied by a decoder responsible for reconstructing the parameters from these latent representations. Considering the memory limitation, the parameter generation is proposed to synthesize latent representations from random noise via a diffusion model by default. After training, we utilize the trained DiffNet to generate the novel parameters as the following chain: random noise $\rightarrow$ reverse process $\rightarrow$ diffused representations $\rightarrow$ pretrained decoder $\rightarrow$ generated parameters.

## 2.3 PARAMETER AUTOENCODER

In order to learn the distribution of high-performing model parameters, we collect the high-performing parameters via train each specific architecture $K$ times and save a set of models' parameters $S = [s_1, \ldots, s_k, \ldots, s_K]$. $K$ denotes the number of these models. We then flatten these parameters into vectors $V = [v_1, \ldots, v_k, \ldots, v_K]$, where $V \in \mathbb{R}^{K \times D}$ and $D$ is the size of the model parameters. After that, an encoder-decoder is utilized to reconstruct these parameters $V$ as the normal autoencoder training. The encoding and decoding processes can be formulated as,

$$Z = [z_1^0, \ldots, z_k^0, \ldots, z_K^0] = \underbrace{f_{\text{encoder}}(V, \sigma)}_{encoding}; V' = [v_1', \cdots, v_k', \cdots, v_K'] = \underbrace{f_{\text{decoder}}(Z, \rho)}_{decoding}, \quad (4)$$

where $f_{\text{encoder}}(\cdot, \cdot)$ and $f_{\text{decoder}}(\cdot, \cdot)$ denote the encoder and decoder that parameterized by $\sigma$ and $\rho$, respectively. $Z$ represents the latent representations and $V'$ is the reconstructed parameters. We default to use a standard autoencoder with a 4-layer encoder and decoder. We study the influences of depth of the encoder and decoder in the Appendix. C.1. Similar to the standard auto-encoder, we

minimize the mean square error (MSE) loss as follows,

$$L_{\mathrm{MSE}} = \frac{1}{K} \sum_1^K \|v_k - v_k'\|^2,$$

(5)

where $v_k'$ is the reconstructed parameters of $k$-th model.

## 2.4 PARAMETER GENERATION

One of the most simple strategies is to directly diffuse the original parameters $V$ to generate the novel parameters $V'$. However, the memory cost of this operation is too heavy, especially when the dimension of $V$ is ultra-large. Based on this consideration, we apply the diffusion model to the latent representations by default. For $Z = [z_1^0, \cdots, z_k^0, \cdots, z_K^0]$, we use the optimization of DDPM (Ho et al., 2020) as follows,

$$\theta \leftarrow \theta - \nabla_\theta \|\epsilon - \epsilon_\theta(\sqrt{\overline{\alpha}_t} x_0 + \sqrt{1 - \overline{\alpha}_t}\epsilon, t)\|^2,$$

(6)

where $t$ is uniform between 1 and $T$, the sequence of hyperparameters $\overline{\alpha}_t$ that indicates the noise strength at each step, $\epsilon$ is the added Gaussian noise, $\epsilon_\theta(\cdot)$ denotes the denoising network that parameterized by $\theta$. After finishing the training of the parameter generation, we directly fed random noise into the reverse process and the trained decoder to generate novel and high-performing parameters.

**Structure Design.** Neural network parameters and image pixels exhibit significant disparities in several key aspects, including data type, dimensions, range, and physical interpretation. Therefore, we introduce specific designs in our approach. Firstly, in contrast to images, neural network parameters mostly have no spatial relevance, so we replace 2D convolutions with 1D convolutions in our diffusion model. Secondly, since batch normalization (BN) operates in the batch dimension and introduces undesired correlations among model parameters, we prefer to utilize channel-wise normalization, such as instance normalization (IN), layer normalization (LN), and group normalization (GN), and explore their characteristic in Sec. 4. The architecture of diffusion is a U-Net that includes an encoder with $D_{\mathrm{enc}}$ layer and a decoder with $D_{\mathrm{dec}}$ layer. More details can be found in the Appendix.

## 3 EXPERIMENTS

### 3.1 SETUP

**Datasets.** We evaluate our proposed method across a wide range of datasets. MNIST (LeCun et al., 1998), CIFAR-10/100 (Krizhevsky et al., 2009), ImageNet-1K. (Deng et al., 2009), STL-10 (Coates et al., 2011), Flowers (Nilsback & Zisserman, 2008), Pets (Parkhi et al., 2012), F-101 (Bossard et al., 2014) are used to study the effectiveness of our method on image classification tasks.

**Architectures.** In the early stage of our exploration, we design three simple architectures that consist of convolutional, pooling, and fully-connected layers, namely S-1, S-2, and S-3. We provide the detailed structures of these three models in the Appendix. We also conduct experiments on classical architectures, such as ConvNet-3 (Gidaris & Komodakis, 2018) (3 convolutional layers and one linear layer.), ResNet-18/50 (He et al., 2016), ViT-Tiny/Base Dosovitskiy et al. (2020), and ConvNeXt-T/B (Liu et al., 2022).

**Training data preparation.** We default to train 200 individual high-performing parameters for each architecture as the training data for DiffNet. For the small models, we obtain these parameters via training from scratch. For the larger models, especially on large datasets, such as ConvNeXt-T on ImageNet-1K, we finetune the released model from timm[1] and densely save the checkpoints.

**Training details.** We first train the parameter autoencoder module for 2000 epochs by default. The latent representations and the parameters of the decoder in the last epoch are saved for the following parameter generation module. Then, we train a diffusion model to generate the diffused representations. The structure of the diffusion model consists of 1D CNNs-based U-Net. We set the learning rate and weight decay as 0.001 and 2e-6 for both modules. We diffuse the full parameter of S-1 to 3 and ConvNet-3. For the memory cost consideration, we only select partial parameters from

---

[1]https://github.com/huggingface/pytorch-image-models

large architectures for diffusing, such as ResNet-18/50, ViT-Tiny/Base, and ConvNeXt-T/B. The last two normalization parameters are set for diffusing by default.

**Inference details.** Given 100 random noise, we fed them into DiffNet to generate 100 models. We then select a model with the best performance on the training set. Finally, we test its accuracy on the validation set and report it for comparison. Model analysis can be found in the Appendix B.1.

Table 1: The best results of the best baseline, ensemble, and ours are shown in the '**baseline/ensemble/DiffNet**' manner. We obtain lossless or even higher performance than the baseline, which demonstrates that DiffNet is able to generate high-performing parameters from random noise. **Bold entries** are best results. The details of S-1, S-2, and S-3 in the Appendix B.1.

| Network\Dataset | MNIST | CIFAR-10 | CIFAR-100 | STL-10 | Flowers | Pets | F-101 | ImageNet-1K |
|---|---|---|---|---|---|---|---|---|
| S-1 | 87.4/87.4/**88.4** | 26.5/26.6/**27.3** | 7.6/7.7/**8.0** | 37.0/37.0/**37.1** | 7.4/7.4/7.4 | 7.5/7.5/**7.7** | 2.6/2.6/**2.7** | - |
| S-2 | 94.7/94.6/**95.0** | 48.0/48.2/**49.2** | 13.6/13.7/13.7 | 48.4/48.4/**48.5** | 10.2/10.2/**10.3** | 10.0/10.1/10.0 | 6.6/6.6/**6.7** | - |
| S-3 | 96.7/96.7/96.7 | 55.7/55.7/**56.0** | 23.6/23.6/**23.8** | 53.4/53.4/**53.5** | 15.5/15.5/15.5 | 11.3/11.3/**11.4** | 7.5/7.5/7.5 | - |
| ConvNet-3 | 99.2/99.2/99.2 | 77.2/77.3/**77.5** | 57.2/57.2/**57.3** | 56.4/56.4/56.4 | 9.8/9.9/**10.3** | 9.5/9.4/**9.8** | 10.3/10.1/**10.4** | - |
| ResNet-18 | 99.2/99.2/**99.3** | 92.5/92.5/**92.7** | 74.5/74.5/**74.7** | 75.5/75.5/75.4 | 49.1/49.1/**49.7** | 60.9/60.8/**61.1** | 71.2/71.3/71.3 | 78.7/78.7/78.7 |
| ResNet-50 | 99.4/99.3/99.4 | 91.3/91.4/91.3 | 71.6/71.6/**71.7** | 69.2/69.1/69.2 | 33.7/33.9/**38.1** | 58.0/58.0/58.0 | 68.6/68.5/68.6 | 79.2/79.2/**79.3** |
| ViT-Tiny | 99.5/99.5/99.5 | 96.8/96.8/96.8 | 86.7/86.8/86.7 | 97.3/97.3/97.3 | 87.5/87.5/87.5 | 89.3/89.3/89.3 | 78.5/78.4/78.5 | 73.7/73.7/**74.1** |
| ViT-Base | 99.5/99.4/99.5 | 98.7/98.7/98.7 | 91.5/91.4/**91.7** | 99.1/99.0/**99.2** | 98.3/98.3/98.3 | 91.6/91.5/**91.7** | 83.4/83.4/83.4 | 84.5/84.5/**84.7** |
| ConvNeXt-T | 99.3/99.4/99.3 | 97.6/97.6/97.7 | 87.0/87.0/**87.1** | 98.2/98.0/98.2 | 70.0/70.0/**70.5** | 92.9/92.8/**93.0** | 76.1/76.1/**76.2** | 82.1/82.1/**82.3** |
| ConvNeXt-B | 99.3/99.3/**99.4** | 98.1/98.1/98.1 | 88.3/88.4/**88.4** | 98.8/98.8/**98.9** | 88.4/88.4/**88.5** | 94.1/94.0/94.1 | 81.4/81.4/**81.6** | 83.8/83.7/**83.9** |

## 3.2 RESULTS

We investigate the performances of our method across eight datasets. Since we default to utilize 200 input models to train the DiffNet, the best performances of these models are reported for fair comparison. We also report the ensemble (simply average) results of the input models. Based on the results in Tab. 1, we have several observations as follows: i). Our method achieves lossless or better results than baseline and ensemble in most cases, which demonstrates that DiffNet can efficiently learn the distribution of high-performing parameters and generate superior models from random noise. ii). DiffNet performs well on various datasets, which indicates the good generality of our method.

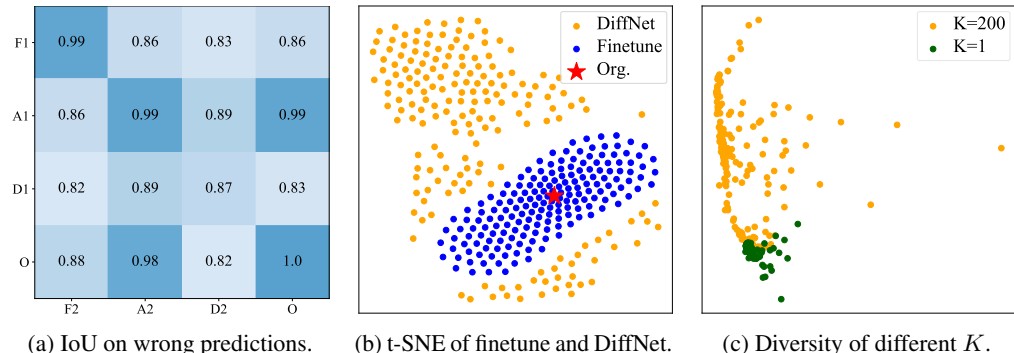

(a) IoU on wrong predictions.  (b) t-SNE of finetune and DiffNet.  (c) Diversity of different $K$.

Figure 3: **DiffNet can generate more diverse high-performing parameters than finetune and add noise.** (a) shows the IoU matrix (defined it in Eq. 7) on wrong predictions. 'F1/2', 'A1/2', and 'D1/2' denote the finetune, add-noise, and DiffNet models are implemented on the same original model 'O' twice independently. (b) presents the t-SNE of parameters distribution of the finetune and DiffNet. (c) studies the diversity of parameters under different numbers ($K$) of the input model. We show the parameters of the BN in ResNet-18 via t-SNE (Van der Maaten & Hinton, 2008b) for visualization.

## 3.3 THE DIVERSITY OF GENERATED PARAMETERS

The diversity of generated parameters is an important metric to verify whether DiffNet actually learns the distribution of high-performing parameters or just memorizes the samples ( *i.e* overfits on samples). We conduct experiments on CIFAR-100 (Krizhevsky et al., 2009) with ResNet-18 (He et al., 2016) under sub-parameter diffusion setting, *i.e.* only diffusing the last two-layer BN. To make a fair comparison, we select two diffused (D1/2), two finetune (F1/2), and two add-noise (A1/2) models with the same accuracy, *i.e.* 74.0% on CIFAR-100. All these models are obtained from the

same original model, named 'O' or 'Org' in Fig. 3a and 3b. We calculate the IoU on their wrong predictions to evaluate the diversity of these models. Formally, the IoU can be written as follows,

$$\text{IoU} = |P_1^{\text{wrong}} \cap P_2^{\text{wrong}}|/|P_1^{\text{wrong}} \cup P_2^{\text{wrong}}|, \tag{7}$$

where $P_{\cdot}^{\text{wrong}}$ denotes the indexes of wrong predictions, $\cap$ and $\cup$ represent union and intersection.

As shown in Fig. 3a, the predictions of the models generated by DiffNet are more diverse than finetune and add noise operations. The IoU of the two models generated by DiffNet is 0.87, while the IoU of the two finetune and add-noise operations are both 0.99. We also compare the IoU between these three operations and the original model. One can find that the DiffNet has the largest contrast with the original models, *i.e.* 0.82 and 0.83 IoU between diffused models and the original one.

In addition to predictions, we also assess the parameter distributions using t-SNE (Van der Maaten & Hinton, 2008a). To mitigate the influence of significant performance differences on visualization, we select 200 models from both finetune and DiffNet with performances better than 74.0%. As shown in Fig. 3b, the parameters generated by DiffNet are obviously more diverse than the model parameters obtained by finetune. That shows our approach learns the distribution of high-performing parameters well. We explore the impact of $K$ in Fig. 3c. It is evident that as $K$ increases, the diversity of generated models also grows. That demonstrates that large K helps to learn the distribution of high-performing parameters. More visualizations are provided in the Appendix.

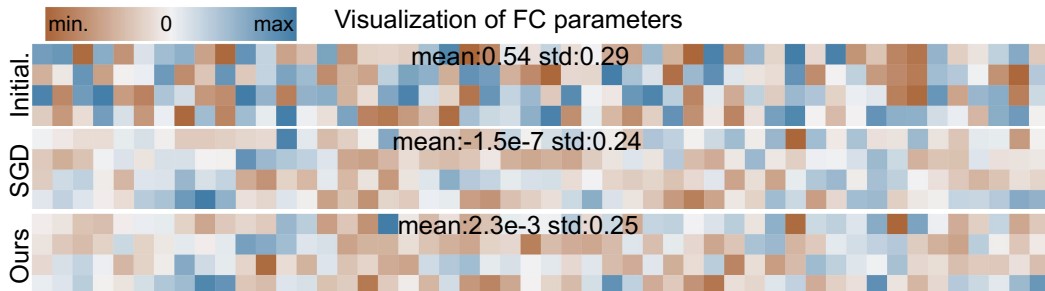

Figure 4: Visualizing the parameters of random initialization, SGD, and ours. DiffNet learns the pattern of high-performing parameters well. More visualizations are provided in the Appendix.

**Visualization.** To provide a better understanding, we compare the parameters generated by DiffNet, SGD, and randomly initialized parameters. Taking the parameters of S-1 as an example, we report the mean and std of the FC parameters in Fig. 4. It is evident that there is a significant difference between the parameters generated by DiffNet and the initialized parameters, mean: 0.54 vs 2.3e-3, std: 0.29 vs 0.25. This visualization strongly confirms that DiffNet can learn the patterns of high-performance parameters and generate good models from random noise.

## 3.4 EFFICIENCY

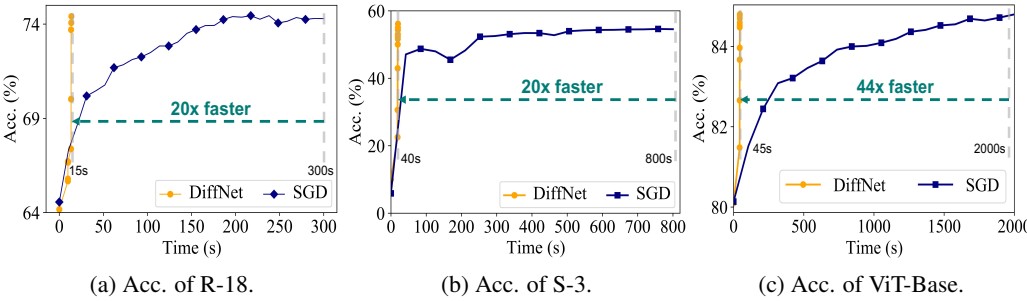

Figure 5: We compare the accuracy curves of our method and SGD under three cases. (a): ResNet-18 on CIFAR-100. (b): S-3 on CIFAR-10. (c): ViT-Base on ImageNet-1K. Our approach speeds up at least 20 × than standard SGD.

To evaluate the efficiency of our method, we compare the accuracy curves of the DiffNet and standard SGD among three cases: 1. sub-parameter diffusion with ResNet-18 on CIFAR-100; 2. full-parameter diffusion with S-3 on CIFAR-10; 3. sub-parameter diffusion with ViT-Base on ImageNet-1K. To make a fair comparison, we utilize the random initialized parameters for DiffNet and SGD. As illustrated in Fig. 5, our method can speed up at least 20 × than the SGD. For example, in Fig. 5a, we just need 15 seconds to achieve 74.7% performance on CIFAR-100, while SGD needs 300 seconds to obtain 74.5%. Notably, on ImageNet-1K, we can speed up by 44 × when compared to the vanilla SGD optimization, which illustrates the more significant promotion when the training dataset is large.

## 4 ABLATION STUDIES AND ANALYSIS

Extensive ablation studies are conducted in this section to illustrate the characteristics of our method. We default to train ResNet-18 on CIFAR-100 and report the best accuracy (if not otherwise stated).

**Latent diffusion vs paramater diffusion.** Autoencoder (AE) is proposed for memory cost consideration and making neural network diffusion possible on large architectures. We explore its effectiveness on S-1, S-2, S-3, ConvNet-3 (C-3), and ConvNet-4 (C-4). The best results of these architectures are reported in Tab. 2a. In the cases from S-1 to S-3, it is evident that whether we use an AE or not, the results are quite similar. Nevertheless, directly diffusing (*i.e.* without AE) the parameters of C-3 and C-4 results in out-of-memory (OOM), even when utilizing a batch size of 1, on a 40G A100 GPU. With AE, DiffNet works well and achieves better results than the original model on C-3 and C-4.

**The number of training models.** Tab. 2b varies the size of training data, *i.e.* the input models number $K$. The sufficient number of input models is important for the performance stability of the generated parameters. This can be explained by the learning principle of the diffusion model: the distribution of the high-performing parameters is hard to learn from a few training samples. However, based on the best results in Tab. 2b, one can find that the performance gap in $K = 1$ and $K = 200$ is very small, *i.e.* 74.5% vs 74.7%.

To comprehensively compare the influences in stability of different $K$, we also calculate the average (avg.) and median (med.) accuracy as metrics of stability of 100 randomly generated models. The stability of generated models in the $K = 1$ setting is much worse than $K = 200$. Specifically, $K = 200$ outperforms $K = 1$ with 5.5% and 3.3% under avg. and med., respectively. Based on the results in Tab. 2b and Fig. 3c, we empirically find that the diffusion model may be hard to model the distribution of the high-performing parameters well if there single input model is used for DiffNet.

$\mathbf{D}_{\text{enc}}$ **and** $\mathbf{D}_{\text{dec}}$. Normally, as the diffusion networks' depth increases, the time of training and inference becomes heavier. Therefore, we study the depth of the encoder and decoder in the diffusion U-Net. The best performance and inference time are reported in Tab. 2c. Note that, the time is the total time of 100 inferences. The default 4-layer autoencoder achieves the highest performance and acceptable inference time, *i.e.* 0.019 seconds per model. We test our DiffNet in a single A100 40G GPU and other detailed hardware information can be found in the Appendix.

**The width of the encoder.** In Tab. 2d, we study the encoder width (number of channels) in the parameter autoencoder module. We report the performance and memory cost with 1, 4, 8, 16, and 32 channels. The channel corresponds to the dimension of $|Z|$ and it is also shown in Tab. 2d. Based on the results, we have several findings: i). The too-narrow encoder can not perform well, which demonstrates the representation ability is poor when the encoder width is very small. ii). As the channels increase from 4 to 16, the best and average performances improve accordingly. That means increasing the width of the encoder can strengthen the performance stability of generated models. iii). If we further increase the channel from 16 to 32, the average performance degrades significantly, *i.e.* 58.5% vs 39.7%. This can be explained by the too-wide encoder being hard for optimization. Considering the performance stability and memory efficiency, we utilize 16 channels by default.

**Normalization layers.** Considering the intrinsic difference between images and neural network parameters, we explore the influence of different normalization strategies. We add the batch normalization (BN), group normalization (GN), and layer normalization (LN) to DiffNet, respectively. We also implement our method without normalization for additional comparison. Their best, average, and median performances of 100 generated models are reported in Tab. 2e. Based on the results, we have the following observations: 1). BN obtains the worst overall performances

Table 2: **DiffNet main ablation experiments.** We design ablations about AE, $K$, network depth, encoder width, normalization, and parameters to diffusion, respectively. If not specified, the default is: using AE, $K = 200$, $D_{enc} = 4$, $D_{dec} = 4$, the encoder width is 16, normalization is IN, and diffuse the deep BN (BN-d) layers. Default settings are marked in gray .

| arch. | org. | w/o AE | w AE |
|---|---|---|---|
| S-1 | 7.6 | 7.8 | **8.0** |
| S-2 | 13.6 | 13.6 | **13.7** |
| S-3 | 23.6 | 23.7 | **23.8** |
| C-3 | 57.2 | oom | **57.3** |
| C-4 | 57.2 | oom | **57.4** |

(a) AE makes DiffNet possible on large architectures.

| $K$ | best | avg. | med. |
|---|---|---|---|
| 1 | 74.5 | 53.0 | 69.7 |
| 10 | 74.6 | 53.1 | 70.6 |
| 50 | 74.6 | 53.3 | 72.0 |
| 200 | **74.7** | **58.5** | 73.0 |
| 500 | 74.6 | 58.4 | **73.1** |

(b) Large $K$ can improve the stability of DiffNet.

| $D_{enc}$ | $D_{dec}$ | best | time (s) |
|---|---|---|---|
| 1 | 1 | 73.4 | 15.2 |
| 2 | 2 | 74.4 | 15.9 |
| 3 | 3 | 74.4 | 16.8 |
| 4 | 4 | **74.7** | 19.4 |
| 5 | 5 | 74.6 | 19.8 |

(c) 4-layer autoencoder in U-Net performs the best accuracy.

| C/$|Z|$ | best | avg. | memory |
|---|---|---|---|
| 1/8 | 74.4 | 48.4 | 1857 |
| 4/32 | 74.5 | 52.0 | 1860 |
| 8/64 | 74.6 | 55.1 | 1863 |
| 16/128 | **74.7** | **58.5** | 1867 |
| 32/256 | 74.6 | 39.7 | 2049 |

(d) 16 channels perform the best trade-off between acc. and mem.

| norm | best | avg | med. |
|---|---|---|---|
| w/o norm | 67.9 | 11.6 | 1.7 |
| BN | 61.1 | 12.4 | 4.2 |
| GN | 74.6 | **67.8** | **74.5** |
| LN | 74.6 | 53.2 | 67.0 |
| IN | **74.7** | 58.5 | 73.0 |

(e) GN has the good acc. but poor diversity. See it in the analysis.

| para. | best | avg. | med. |
|---|---|---|---|
| FC | 74.5 | 47.0 | 37.5 |
| BN-s | 74.1 | **68.1** | **73.4** |
| BN-d | **74.7** | 58.5 | 73.0 |
| Conv-s | 72.9 | 67.5 | 69.5 |
| Conv-d | 73.9 | 38.9 | 61.4 |

(f) DiffNet works better on deep (-d) layers than shallow (-s) layers.

on all three metrics, which aligns well with our analysis in Sec. 2.4. 2). LN and GN perform better than without normalization, *i.e.* 'w/o norm' in the Tab. 2e. It could be explained by some outlier parameters affect the performance a lot. 3). From the metrics, we find that GN is the best normalization. However, the diversity of the generated models of GN is obviously poor when compared to IN (see the **lower left** wrap figure on page 7).

**Where to apply DiffNet.** To save the data preparation cost, we default to diffuse the parameters of the last two normalization layers. We also explore the impact of diffusing the shallow layers and other types of parameters, such as convolutional and fully-connected parameters. As shown in Tab. 2f, we empirically find that diffusing the deep layers can obtain better best accuracy than diffusing the shallow layers. That is because generating shallow-layer parameters is more likely to accumulate errors in the forward and backward processes compared to generating deep-layer parameters. Another finding is diffusing BN parameters performs better than other parameters. We show the performance comparisons of applying DiffNet in all BN layers of ReNet-18 on CIFAR-100 in the Appendix.

**Analyzing the Interchangeability of SGD and DiffNet.** Considering the similarity between SGD and diffusion steps, we design experiments to explore the interchangeability of these two technologies. Specifically, we test the interchangeability in two cases. The first one: we explore the performance of models generated by DiffNet when they continue to be trained with SGD in Fig. 6a. The second one: we also investigate the performance of introducing DiffNet into models that have been trained with SGD for some iterations in Fig. 6b. All results are obtained from training CIFAR-100 with ResNet-18 under the sub-parameter diffusion setting. The experimental results demonstrate the good interchangeability between SGD and DiffNet. Another benefit is that combining DiffNet and SGD can improve training efficiency compared to utilizing pure SGD. For example, SGD needs 300s (see it in Fig. 5a) to obtain a high-performing model, while we only need 100s and 8s to finish it in Fig. 6b and 6a, respectively. We conduct more detailed experiments in Fig. 9 and 11 in the Appendix.

## 5    RELATED WORK

**Diffusion models** have achieved remarkable results in visual generation. These methods (Ho et al., 2020; Dhariwal & Nichol, 2021; Ho et al., 2022; Peebles & Xie, 2022; Hertz et al., 2023; Li et al., 2023) are based on non-equilibrium thermodynamics (Jarzynski, 1997; Sohl-Dickstein et al., 2015), and the its pathway is similar to GAN (Zhu et al., 2017; Isola et al., 2017; Brock et al., 2018a; Karras et al., 2019), VAE (Kingma & Welling, 2013; Razavi et al., 2019), and flow-based model (Dinh et al., 2014; Rezende & Mohamed, 2015). Diffusion models can be categorized into

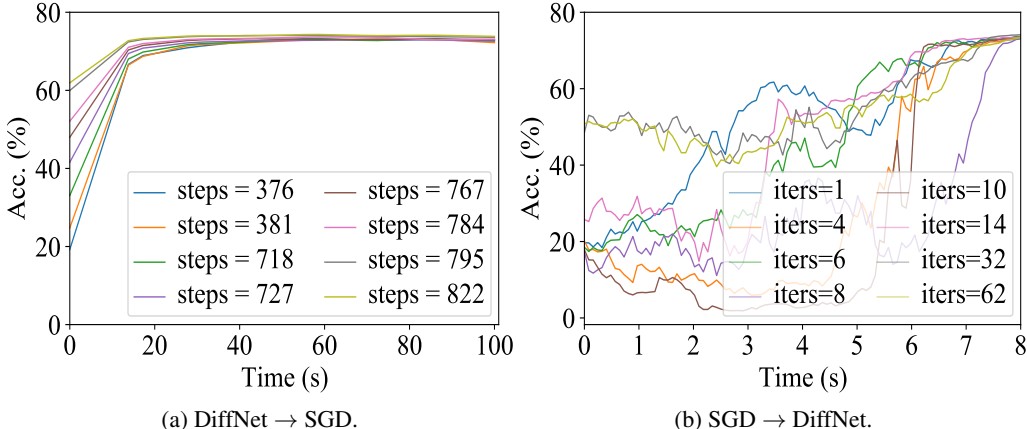

(a) DiffNet → SGD.

(b) SGD → DiffNet.

Figure 6: (a) evaluates the effect of continuing the SGD optimization from the parameters obtained by DiffNet with different time steps (selected the ACC. between 20%~60%). (b) shows the performance curves of adding DiffNet (last 100 steps) to the parameters that are optimized by different SGD iterations. (a) and (b) consistently demonstrates the good interchangeability of SGD and DiffNet.

three main branches. The first branch focuses on enhancing the synthesis quality of diffusion models, exemplified by models like DALL·E 2 (Ramesh et al., 2022), Imagen (Saharia et al., 2022), and Stable Diffusion (Rombach et al., 2022). The second branch aims to improve the sampling speed of diffusion models, including DDIM (Song et al., 2021), Analytic-DPM (Bao et al., 2022), and DPM-Solver (Lu et al., 2022). The final branch involves reevaluating diffusion models from a continuous perspective rather than a discrete one, like score-based models (Song & Ermon, 2019; Feng et al., 2023).

**Parameter Generation**   is a key problem in the deep learning area. Stochastic neural networks (SNNs) (Bottou et al., 1991; Wong, 1991; Schmidt et al., 1992; Murata et al., 1994; Sompolinsky et al., 1988) introduce randomness to improve the robustness and generalization capabilities of neural networks. HyperNet (Ha et al., 2017) dynamically generates the weights of a main model with variable architecture. Smash (Brock et al., 2018b) introduces a flexible scheme based on memory read-writes that can define a diverse range of architectures. Peebles et al. (2023) collect 23 million checkpoints and train a conditional parameters generator via a transformer-based diffusion model. MetaDiff (Zhang & Yu, 2023) introduces a diffusion-based meta-learning method for few-shot learning, where a layer is replaced by a diffusion U-Net (Ronneberger et al., 2015). HyperDiffusion (Erkoç et al., 2023) directly utilizes a diffusion model on MLPs to generate new neural implicit fields. Different from them, we analyze the intrinsic differences between images and parameters and design corresponding modules to learn the distributions of the high-performing parameters.

## 6  Discussion and Conclusion

High-performing parameters are fundamental to deep learning. Previous works propose several learning paradigms, including supervised learning (Krizhevsky et al., 2012; Simonyan & Zisserman, 2014; He et al., 2016; Dosovitskiy et al., 2020), self-supervised learning (Devlin et al., 2018; Brown et al., 2020; He et al., 2020; 2022), and so on. In this study, we observe diffusion models can also be used to generate high-performing neural network parameters and show their superiority. Employing diffusion steps for parameter updates represents a potentially novel paradigm in deep learning.

On the other hand, we note that images/videos and parameters are signals of a different nature, and this difference must be addressed carefully. Moreover, while diffusion models have achieved considerable success in image/video generation, their exploration in the context of parameters is limited. These lead to a series of challenges for neural network diffusion. We propose a preliminary approach to tackle some of these challenges. However, there are still unresolved challenges, including memory constraints for generating parameters of large architectures under the full-parameter diffusion setting, the need for more reasonable structure design, and ensuring the performance stability of generated parameters. We hope this study inspires future work. The code will be made publicly available.

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

## A DATASETS AND IMPLEMENTATION DETAILS

In this part, we show the detailed introductions of used datasets and implementations.

**Datasets** We evaluate the effectiveness of DiffNet on 8 datasets. To be specific, **CIFAR-10/100** (Krizhevsky et al., 2009). The CIFAR datasets comprise colored natural images of dimensions $32 \times 32$, categorized into 10 and 100 classes, respectively. Each dataset consists of 50,000 images for training and 10,000 images for testing. **ImageNet-1K.** (Deng et al., 2009) derived from the larger ImageNet-21K dataset, ImageNet-1K is a curated subset featuring 1,000 categories. It encompasses 1,281,167 training images and 50,000 validation images. **STL-10** (Coates et al., 2011) comprises $96 \times 96$ color images, spanning 10 different object categories. It serves as a versatile resource for various computer vision tasks, including image classification and object recognition. **Flowers** (Nilsback & Zisserman, 2008) is a dataset comprising 102 distinct flower categories, with each category representing a commonly occurring flower species found in the United Kingdom. **Pets** (Parkhi et al., 2012) includes around 7000 images with 37 categories. The images have large variations in scale, pose, and lighting. **F-101** (Bossard et al., 2014) consists of 365K images that are crawled from Google, Bing, Yelp, and TripAdvisor using the Food-101 taxonomy.

**Implementation details.** All experiments are conducted with an A100 GPU server. The CPU cores of our server are 298. The training dataset comprises 200 fine-tuned models, with each epoch preserving one model in the case of the small-scale dataset (MNIST, CIFAR etc.) and retaining 100 models during a single epoch in the context of the large-scale dataset (ImageNet-1K). A total of 500,000 epochs were conducted, with the initial 50,000 epochs only the AE module is trained, and subsequently, the Diffusion module undergoes training. For DiffNet, default learning rate of AE and diffusion model is set to 0.01, and the weight decay is also set to the default value of 0.0005.

## B EXPLORATION PROCESS

In this part, we explain the exploration process of the neural network diffusion, including the details of structure design, hyperparameters, and related ablations. The defaulted settings are mainly based on the results of explorations.

### B.1 HAND-DESIGNED SMALL MODELS

We first explore our method on very small architectures and simple datasets, such as MNIST and CIFAR-10. Here, we show the details of the small architectures as follows.

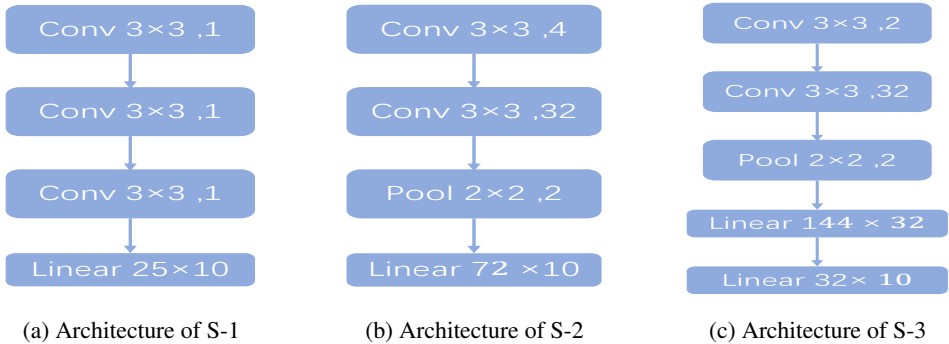

(a) Architecture of S-1      (b) Architecture of S-2      (c) Architecture of S-3

Figure 7: Detailed structures of S-1, S-2, and S-3. We begin our explorations based one these small architectures.

The number of parameters of these designed small architectures are 227, 1066, and 5066. S-1 consists of 3 convolutional layers with 1-channel $3 \times 3$ kernel and one linear layer. We increase the number of channels and add a pooling layer in S-2. We design two cascaded linear layers in S-3. We then follow the famous diffusion model, DDPM (Ho et al., 2020), to build our training code. DDPM mainly includes a U-net that consists of an encoder and decoder. DDPM aims to generate high-quality

images via a 2D convolutional-based U-Net. Batch Normalization (BN) has achieved remarkable results in many tasks. We directly utilize the structure of DDPM with BN and show results in Tab. 3.

Table 3: Using 1D CNNs with IN performs better than 2D CNNs with BN.

| Method | Dataset | Time (s) | Best | Average | Median | Worst | Memory (MB) |
|---|---|---|---|---|---|---|---|
| BN+2D CNNs | MNIST | 16 | 79.7 | 53.2 | 60.2 | 38.9 | 1089 |
| IN+1D CNNs | MNIST | 13 | 99.3 | 98.7 | 99.5 | 64.5 | 1058 |
| BN+2D CNNs | CIFAR-10 | 16 | 77.5 | 57.6 | 57.1 | 45.6 | 1423 |
| IN+1D CNNs | CIFAR-10 | 14 | 92.7 | 78.6 | 87.6 | 45.4 | 1326 |
| BN+2D CNNs | CIFAR-100 | 18 | 19.5 | 19.5 | 19.5 | 19.5 | 1976 |
| IN+1D CNNs | CIFAR-100 | 17 | 74.7 | 58.5 | 73.0 | 32.5 | 1867 |

As shown in Tab. 3, one can find the BN and 2D CNNs do not work very well for parameter generation. We analyze it from the following aspects: 1). Parameters are not like pixels in vision data. Images have local continuity and similarity, but the parameters may not have. Therefore, directly applying 2D CNNs in parameter generation may not be reasonable. 2). BN is originally proposed to speed up the training and model the general distribution of images, *i.e.* some common patterns. However, the parameters of different checkpoints are individual.

In our work, we treat network parameters as sequential data. We replace the 2D convolutions with 1D convolutions. Empirically, we have found that this approach yields good performance. We compare their performances in the Tab. 3. The experimental results on three small models consistently demonstrate the effectiveness of 1D convolutions and instance normalization (IN). Unless otherwise specified, we default to using 1D convolutions and IN.

**Why 1D CNNs?** Here naturally raises a question: are there alternatives to 1D convolutions? The answer is yes, we can use pure fully-connected (FC) layers as an alternative. Therefore, we compared the performance of FC layers and 1D convolutions. Based on our experimental results in Tab. 4, 1D CNNs consistently outperform FC in all architectures. Meanwhile, the memory occupancy of 1D CNNs is smaller than FC. Considering the performance and efficiency, we default to use 1D CNNs.

Table 4: Comparison of using 1D CNNs and FC. 1D CNNs perform better than FC, especially in memory and time.

| Arch. | Method | Dataset | Time (s) | Best | Average | Median | Worst | Memory (MB) |
|---|---|---|---|---|---|---|---|---|
| S-1 | FC | MNIST | 12 | 87.4 | 67.8 | 77.5 | 55.4 | 1077 |
| S-1 | 1D CNNs | MNIST | 11 | 88.4 | 68.7 | 78.3 | 58.7 | 1004 |
| S-2 | FC | MNIST | 15 | 94.4 | 70.6 | 85.5 | 58.3 | 1165 |
| S-2 | 1D CNNs | MNIST | 14 | 95.0 | 72.5 | 84.5 | 59.2 | 1058 |
| S-3 | FC | MNIST | 15 | 96.5 | 87.1 | 95.4 | 10.0 | 1233 |
| S-3 | 1D CNNs | MNIST | 14 | 96.7 | 90.3 | 92.1 | 66.8 | 1122 |
| ConvNet-3 | FC | MNIST | 17 | 98.0 | 90.1 | 93.6 | 70.2 | 1375 |
| ConvNet-3 | 1D CNNs | MNIST | 16 | 99.2 | 92.1 | 94.2 | 73.6 | 1244 |

**Abalation of $K$.** In the early stages of our exploration, we aim to make the diffusion model to learn the distribution of high-performance model parameters. To achieve this, we initially trained 10,000 small models individually. However, training such a large number of models is extremely time-consuming, and the issue of redundancy becomes quite prominent. Therefore, we conduct experiments to evaluate the impact of different values of $K$ on the performance and stability of the generated models. As shown in Tab. 5, The experimental results (test on the MNIST dataset) indicate that once $K$ exceeds 200, the performance of DiffNet reaches a stable state. Taking efficiency into consideration, we set the default value of $K$ to 200.

**How to select the generated parameters?** As mentioned above, DiffNet can rapidly generate numerous high-performance models. How do we evaluate these models? There are two primary strategies. The first one is to directly test them on the *validation set* and select the best-performing model. The second one is to compute the loss of model outputs compared to the ground truth on the

Table 5: Increasing the number ($K$) of input models helps the performance stability of generated models.

| $K$ | best | avg. | med. |
|-----|------|------|------|
| 1 | 88.0 | 53.0 | 69.7 |
| 10 | 88.1 | 53.1 | 70.6 |
| 50 | 88.1 | 53.3 | 72.0 |
| 200 | **88.4** | **58.5** | 73.0 |
| 500 | 88.3 | 58.4 | **73.1** |

(a) Evaluation of $K$ on S-1

| $K$ | best | avg. | med. |
|-----|------|------|------|
| 1 | 94.5 | 57.3 | 80.0 |
| 10 | 94.5 | 63.3 | 84.5 |
| 50 | 94.8 | 57.4 | 84.6 |
| 200 | **95.0** | 64.8 | 88.9 |
| 500 | **95.0** | **64.9** | **89.3** |

(b) Evaluation of $K$ on S-2.

| $K$ | best | avg. | med. |
|-----|------|------|------|
| 1 | 96.0 | 40.4 | 20.4 |
| 10 | 96.0 | 43.1 | 26.1 |
| 50 | 96.2 | 45.3 | 26.6 |
| 200 | **96.7** | **47.6** | **33.5** |
| 500 | 96.6 | 46.1 | 32.0 |

(c) Evaluation of $K$ on S-3.

(a) Acc. of R-18 on MNIST. (b) Acc. of S-3 on CIFAR-10. (c) Acc. of ViT-Base on IN-1K.

Figure 8: The training accuracy on MNIST, CIFAR-10, and ImageNet-1K (IN-1K) are highly-consistent with their validation accuracy. We default to select the best-performed model on the training set.

*training set* to choose a model. We generated 100 model parameters and displayed their accuracy curves on both the training and validation sets in Fig. 8. The experimental results indicate that DiffNet exhibits a high level of consistency between the training and validation sets. To provide a fairer comparison with baseline methods, we default to choose the model that performs the best results on the training set and compare it with the baseline.

**Scaling the method to large architectures** We validated the effectiveness of our method on small-parameter models. What happens when we apply our method to models with a large number of parameters? To investigate this, we conducted experiments that demonstrate the time and memory requirements of our method under different parameter settings.

From the experimental data in Tab. 6, we can observe that as the number of parameters increases, our method's training time and memory usage also increase. We encounter an out-of-memory issue when using ConvNet3. This indicates that we need to mitigate this issue when dealing with a large number of parameters. Therefore, inspired by the latent diffusion model (LDM) (Rombach et al., 2022), we propose an AutoEncoder (AE) to reduce the dimension of model parameters into a latent space representation, effectively alleviating the memory overhead while achieving good performance.

## C EXTERNAL EXPERIMENTS

### C.1 THE DEPTH OF ENCODER AND DECODER IN PARAMETER AUTO-ENCODER MODULE

As previously mentioned, we employed a standard autoencoder in the parameter autoencoder module, consisting of a 4-layer encoder and decoder. We conduct experiments to evaluate the impact of different depths of both the encoder and the decoder on performance. From the experimental results in Tab. 7, we have the following findings: 1). Too-shallow encoder or decoder affects the performance significantly, which indicates the shallow auto-encoder can not learn the distribution well. 2). Using a 4-layer autoencoder improvements the overall performance largely, as well as outperforms the original model's performance.

Table 6: Exploring the effectiveness of autoencoder among S-1, 2, 3, and C-3. We report their best, average, and median performances. For efficiency, memory cost and training time are also provided.

| Arch. | best | avg. | med. | mem. | time |
|-------|------|------|------|------|------|
| S-1 | 88.4 | 87.1 | 87.2 | 2765 | 35 |
| S-2 | 95.0 | 94.7 | 94.5 | 3137 | 41 |
| S-3 | 96.2 | 93.3 | 94.9 | 3548 | 45 |
| C-3 | oom | - | - | - | - |

(a) w/o AE on S-1, 2, 3, and C-3.

| Arch. | best | avg. | med. | mem. | time |
|-------|------|------|------|------|------|
| S-1 | 88.4 | 68.7 | 78.3 | 2032 | 30 |
| S-2 | 95.0 | 72.5 | 84.5 | 2278 | 34 |
| S-3 | 96.7 | 90.3 | 92.1 | 2649 | 40 |
| C-3 | 99.2 | 92.1 | 94.2 | 5298 | 63 |

(b) w AE on S-1, 2, 3, and C-3.

Table 7: Exploration the depth of encoder and decoder in parameter autoencoder module.

| Arch. | org. | best | avg. | med. | mem. | time |
|-------|------|------|------|------|------|------|
| 1 | 74.5 | 51.5 | 3.6 | 1.0 | 1786 | 18.0 |
| 2 | 74.5 | 70.0 | 42.3 | 42.0 | 1843 | 18.3 |
| 3 | 74.5 | 69.3 | 23.5 | 14.3 | 1855 | 19.0 |
| 4 | 74.5 | **74.7** | **58.5** | **73.0** | 1867 | 19.4 |

(a) Evaluation of the depth of encoder.

| Arch. | org. | best | avg. | med. | mem. | time |
|-------|------|------|------|------|------|------|
| 1 | 74.5 | 72.5 | 52.1 | 48.7 | 1779 | 17.8 |
| 2 | 74.5 | 72.3 | 53.1 | 50.6 | 1837 | 18.6 |
| 3 | 74.5 | 74.0 | 40.9 | 46.1 | 1859 | 19.1 |
| 4 | 74.5 | **74.7** | **58.5** | **73.0** | 1867 | 19.4 |

(b) Evaluation of the depth of decoder.

## C.2 PERFORMANCES OF DIFFUSING ALL LAYERS OF RESNET-18

In our study, we explore the application of our method to generate certain layers of both deep and shallow variants of the ResNet-18 (R-18) model, including BN (Batch Normalization) and Convolutional layers. To gain a more comprehensive understanding of the characteristics of our method, we further applied diffusion to other batch normalization layers in the R-18 architecture. Subsequently, we conduct an analysis of their performance. As shown in Tab. 8, one can find that diffusing batch normalization performs consistently well.

Table 8: Illustrations of diffusing the different batch normalization layers in ResNet-18. Diffusing the parameters of BN layers performs overall good.

| Acc/layer | layer1.1.bn | layer2.0.bn | layer2.1.bn | layer3.0.bn | layer3.1.bn | layer4.0.bn | layer4.1.bn |
|-----------|-------------|-------------|-------------|-------------|-------------|-------------|-------------|
| Best | 74.2 | 74.1 | 74.5 | 74.5 | 74.6 | 74.5 | 74.7 |
| Avg. | 67.3 | 64.8 | 67.4 | 59.7 | 58.5 | 60.3 | 58.5 |

## C.3 HOW ABOUT DIFFUSING PARAMETERS MORE THAN 1000 STEPS?

During the inference process, we follow DDPm and default to denoise the random noise 100 steps. It came to our attention that not all parameters/representations could achieve excellent performance within the first 1000 diffusion steps. Consequently, we conducted experiments to investigate the impact of increasing the number of diffusion steps on these models. To simplify the problem, we only add the 1000th noise multiple times. We respectively add extra 500, 1000, and 1500 steps. As shown in Fig. 9, longer diffusion steps can improve the quality of the parameters if they are not performing well in the 1000th step.

## D CODE AND VISUALIZATIONS

### D.1 CODE

We have submitted the source code as the supplementary materials in a zipped file named as 'DiffNet.zip' for reproduction. A README file is also included for the instructions for running the code. We will make it public after the submission period.

### D.2 VISUALIZATIONS OF DIFFERENT NORMALIZATION STRATEGIES

We visualize the t-SNE comparison among instance normalization (IN), layer normalization (LN), and group normalization (GN). To mitigate the influences of large performance differences in generated

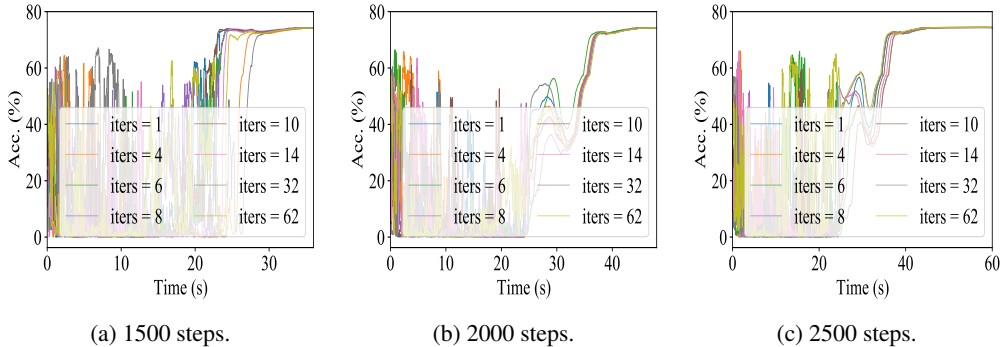

(a) 1500 steps.    (b) 2000 steps.    (c) 2500 steps.

Figure 9: Exploration of adding more than 1000 diffusion steps. Longer diffusion makes the poor models better.

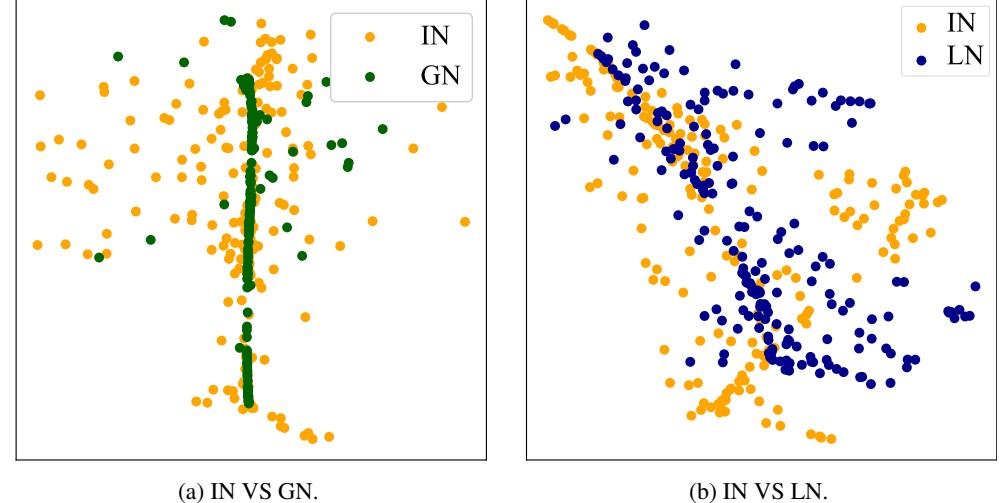

(a) IN VS GN.    (b) IN VS LN.

Figure 10: Comparison of the diversity of different normalization strategies. Considering the poor performance of batch normalization (BN), we just show the results of IN, GN, and LN. Because the performance may lead to influence for visualization.

models, we only show the parameters that perform better than 74.0% on CIFAR-100. As shown in Fig. 10 and Tab. 2e, one can find using IN performs better diversity than GN and better performance than LN. To make our method learn a wide range of high-performing parameter distributions, we default to use IN.

### D.3 EVALUATION OF THE ACCURACY CURVES WITH DIFFERENT DIFFUSION STEPS

Following the experiments in Fig. 6b, we further explore the accuracy curves with different diffusion steps. As shown in Fig. 11, we continue to update the SGD parameters via last 200, 500, and 700 steps. One can find that all these steps can achieve high-performing results. The difference is that as the steps increase, the randomness in the diffusion stage also becomes large. The final results are similar under different steps. It shows our method can perceive the status of the current parameters and adjust it accordingly.

### D.4 PARAMETER PATTERNS VISUALIZATIONS

DiffNet aims to learn the distribution of high-performing parameters. We visualize the parameter patterns of SGD-trained ResNet-18 models. As shown in Fig. 12, we empirically find there exists some patterns among the high-performing parameters. For example, the weights of layer1.0 conv1

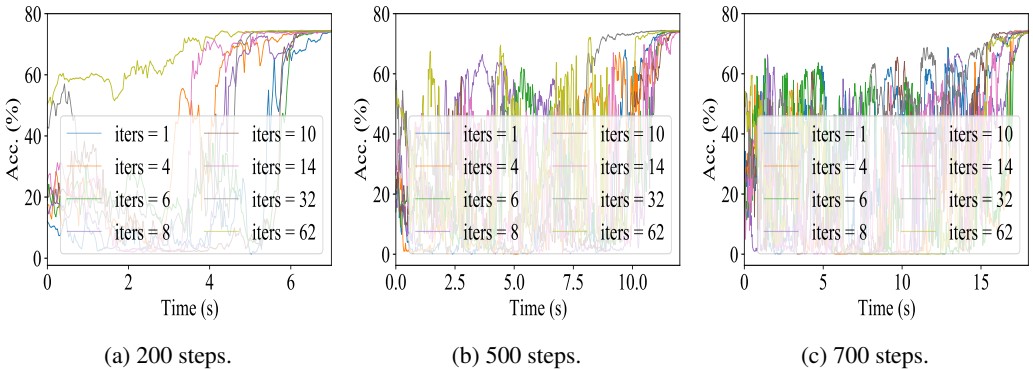

(a) 200 steps.        (b) 500 steps.        (c) 700 steps.

Figure 11: Performance curves of using difference diffusion steps. Given different SGD models, we continue to update the parameters via different numbers of diffusion steps.

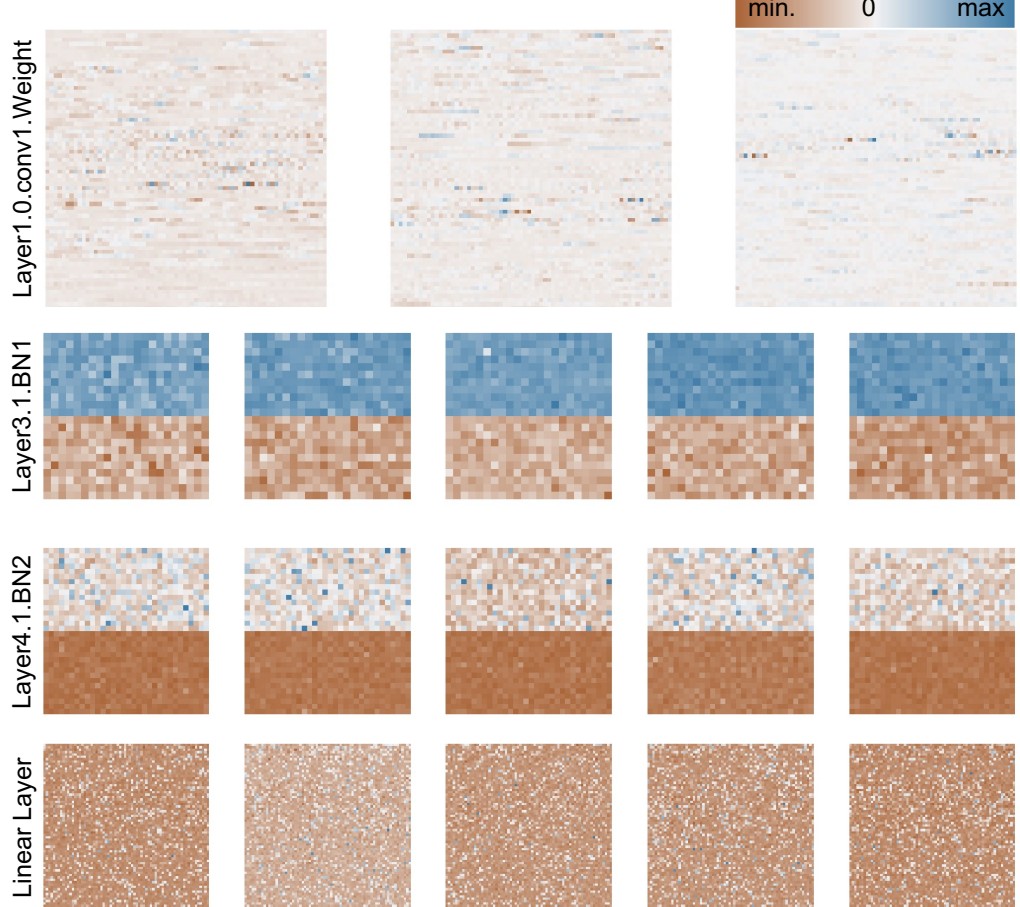

Figure 12: Visualization of the parameters of the convolutional, BN, and linear layers in ResNet-18 SGD-trained models. We show 3 convolutional parameters of layer1.0, 5 BN parameters of layer3.1 and layer4.1, and 5 parameters of linear layer.

have lots of values nearby zero. Note that, we show the weights and biases of BN in one picture, *i.e.* the top part includes weights, while the bottom part includes biases. It is evident that different parameters have their own patterns. These results support the diffusion-based design of our method.

