# OpenReview forum: "Neural Network Diffusion"
_ICLR.cc/2024/Conference — ICLR 2024 Conference Withdrawn Submission_

### Official Review · Reviewer_jEgU · 2023-10-31

**Soundness:** 2 fair
**Presentation:** 1 poor
**Contribution:** 2 fair
**Rating:** 3
**Confidence:** 3

**Summary:**

This paper uses diffusion model to learn and generate network parameters.  To lower the computation cost, the diffusion model is trained in the latent space with the help of an encoder and decoder in the style of stable diffusion.  Empirically, the authors claim to have outperformed SGD-trained models.

**Strengths:**

As far as I know, this is the first work that utilises a diffusion model for this specific task.  The method is clearly explained in the paper.  The writing is easy to follow.

**Weaknesses:**

The main weakness of this paper is the limited novelty of the proposed approach.  IMO, this is just a direct application/adaption of stable diffusion for another application.  In principle, there's nothing new about the proposed approach.

Additionally, I found the presentation of the experimental results confusing.  I will list the questions in the section below.

**Questions:**

1. motivation:  in the introduction, the authors explained that the choice of diffusion models for this task was based on its "capability to transform a given random distribution to a specific one".  Don't all generative models have this capability?  Isn't the fact that diffusion models achieve this by denoising gradually, similar to SGD, actually the drawback of this model?  If possible, we'd prefer generating high performing parameters in one step during inference using other generative models?

2. In table 1, what is the evaluation metric for those scores?

3. What does it mean to have a single data point representing the original model in the parameter distribution in figure 3?

4. How can you tell the proposed method is better than SGD in figure 4?  Is this just evident by having a difference between the generated parameters and the initial ones?  The SGD parameters are also, if not more, different.  Does being more different necessarily mean better? Can't tell from this figure.

5.  It might be better if you took less seconds or provide a zoom-in look in figure 5.

---

### Official Review · Reviewer_GLkA · 2023-10-31

**Soundness:** 3 good
**Presentation:** 3 good
**Contribution:** 2 fair
**Rating:** 3
**Confidence:** 4

**Summary:**

This paper introduces a diffusion-based generative model for neural network parameters (checkpoints). The key distinction from prior works is that the diffusion is run on a latent space, not on the parameter space directly. The weights of a neural network are flattened into a vector and then auto-encoded with standard autoencoder. Then a standard DDPM is applied to the latent space. Once trained, the proposed model coined as DiffNet, can take a random noise to reverse it to a latent variable representing a network, and the decoder can decode it to an actual parameter. The efficiency of the proposed algorithm is verified on a benchmark containing various architectures and image datasets.

**Strengths:**

- The paper is tackling an interesting problem; neural network parameter generation is personally an interesting research topic.
- The paper is fairly well-written and easy to follow. The algorithm is straightforward and the authors provided enough details, so I imagine it to be easily reproducible.
- The paper provides extensive qualitative analysis and ablation study about the proposed method.

**Weaknesses:**

- The novelty is limited. The very ideas of "neural network diffusion" or "using diffusion model for neural network parameters" or even the concept of "treating neural network parameters as an object to define a generative model", are not new. If this paper had been the pioneering source for these ideas, its contribution would have been particularly commendable. The only difference from the previous works is that the diffusion is applied to the latent space, which is also a standard approach for diffusion models.
- No comparison to any existing works.
- To really demonstrate the generality of the presented method, it should be tested on out-of-distribution dataset or architectures; that is, the method should be verified to work for the architectures or datasets that were not included in training. But as far as I could understand all the results were done for a fixed set of architectures and datasets for both training and validation.
- Lack of useful applications. While I value the direction, the experiments conducted in the papers do not remind me of any practical application for which DiffNet plays indispensable roles. For instance, what is the use of having a bunch of similar-performing neural network parameters for a fixed set of architectures, fixed set of datasets, and fixed set of tasks (although the experiments only cover classification task)? Why do we need 100 additional models generated from random noises when we already have a single (or few) competent models?

**Questions:**

- What is the "standard autoencoder architecture"? Is it MLP, convnet, or transformers? Also, it seems that the encoder does not take the symmetries in the neural network weights (e.g., permutation symmetry) into account, which means that multiple neural network parameters defining exactly the same neural networks may be mapped into different latent variables.

- In Table 1, what is the precise meaning of "ensemble"? Is it the accuracy of the "ensemble" of 200 input models? If so, it is weird, because in none of the presented cases the ensemble model showed significant ensemble gain, while it is verified by numerous papers and empirical experimental results that ensembling a moderate number of models leads to a significant gain in the accuracy. If the ensemble (simple average) is simply an average accuracy of 200 input models, then why for some cases (e.g., S-1 CIFAR-10 and S-1 CIFAR-100) the baseline (this is the best performing one, right?) is worse than the ensemble?

---

### Official Review · Reviewer_aG24 · 2023-11-02

**Soundness:** 3 good
**Presentation:** 3 good
**Contribution:** 3 good
**Rating:** 6
**Confidence:** 3

**Summary:**

The paper proposes to learn the parameters of a neural network by diffusion model. Specifically, it builds a latent diffusion model (LDM) where the parameters of the neural networks are encoded into the latent space for diffusion, and the portion of the neural network for diffusion is studied. Experiments are conducted with SGD-trained classification networks of multiple datasets and show promising results.

**Strengths:**

1. The research topic is pretty interesting and novel. Instead of using SGD for neural networks learning, we can use a generative approach to generate the parameters as well.
2. Experiments are conducted on multiple classification datasets with promising results over SGD learning.

**Weaknesses:**

1. I am mainly questionable about the set of parameters that is being generated by the diffusion model for each network architecture.
    - Do the authors try to generate all the parameters of a model? The paper does not mention it.
    - What is the number of parameters that are being generated for each task in Table 1? The paper does not mention it.
    - For VIT architecture, I think many of them do not use batch normalization but instead doing group normalization, instance normalization, etc. Does the approach still work? More discussion into the VIT architecture would be needed.

**Questions:**

Please see the weakness section.